# Exploring Cluster Growth Using a Simple Domino Tiling

**Darren J. Goossens**

School of Physical Environmental and Mathematical Sciences, University of New South Wales, Canberra, ACT 2600, Australia; darren@brittlegum.com.au or d.goossens@adfa.edu.au; Tel.: +61-0474-122-581

**Abstract:** This paper and its deposited material explore clustering of $2 \times 1$ dimers (dominoes) subject to simple interactions and temperature. Much of the work in domino tilings has been statistical, combinatoric and thermodynamic in nature. Instead, here, the domino is used as a simple model of a non-spherical molecule to explore aggregation, rather as if the molecules were interacting in solution. As a result, the work does not look at how many ways there are to tile a plane, but at how the cluster evolves with different parameters in the potential that governs the clustering. These parameters include the rules used to select which of the many possible dominoes will be added to the cluster, and temperature. It is shown that qualitative changes in clustering behaviour occur with temperature, including affects on the shape of the cluster, vacancies and the domain structure.

**Keywords:** domino; dimer; clustering; packing

## 1. Introduction

The field of tiling two-dimensional (2D) space with regular polygons is large and well explored [1–3]. The work presented here looks at how $2 \times 1$ and $1 \times 2$ dominoes will cluster on an "infinite" 2D plane, subject to simple rules that mimic some aspects of the chemistry of a clustering system. The model is to some extent a toy, yet shows a range of interesting behaviours.

The classic model of a crystalline material as a regular grid of unit cells has been extraordinarily successful. Recent times have seen increasing interest in variations away from that idea. Quasicrystals [4–6], modulated structures [7] and disordered structures [8] have become fruitful fields of study.

Quasicrystals are related to Penrose tilings [9], which allow a tiling of the plane that is both non-repeating and point diffractive (the diffraction pattern of the tiling shows sharp features, suggestive of order, while the non-repeating nature of the pattern is suggestive of disorder) [10]. The Penrose tilings use two different "cells" to tile the plane, and these can be arranged with varying orientations. This is very different from a conventional crystal structure in which there is only a single unit cell, and it is always in the same orientation.

A very simple model that allows non-periodic tiling is one in which there is only one type of cell, but it can take on more than one orientation. This is sort of halfway between a conventional structure and a Penrose tiling. A simple example of such a system is a domino, a $2 \times 1$ tile, which can take on one of two orientations, which may be termed vertical and horizontal.

Plainly, it is possible to tile 2D space with an array of dominoes, and the statistics and thermodynamics of this tiling have been explored [11,12]. The array could be periodic with all tiles parallel, or with some fraction vertical and the rest horizontal, or could be non-periodic, although a structure non-periodic in both directions would require both orientations to be present. If only one

orientation is used, a stacking-faulted structure in which strings of dominoes are randomly shifted by half the length of the long side of the cell would be possible.

This work explores the growth of domino clusters subject to various rules that act to control the growth in non-determinative ways. Such a system is related to the clustering of dimers [13,14] and the growth of crystals in solution.

## 2. Preliminaries

The basic conception is that of a 2D "solution" in which dominoes, $2 \times 1$ tiles, are floating about and eventually a cluster nucleates and grows, subject to a simple potential (Figure 1).

The potential used is a local, scalar potential; dominoes with a greater perimeter in common with the existing cluster are the most likely to be added. Figure 2 illustrates this.

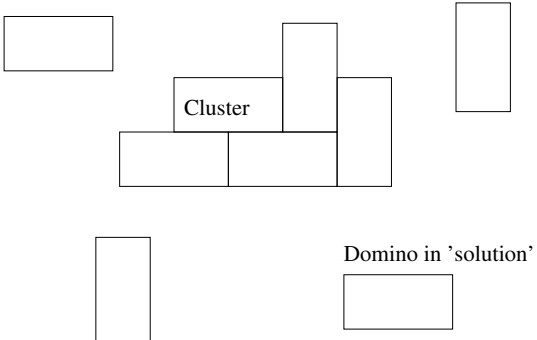

**Figure 1.** The model; a cluster exists, within a 2D solution of dominoes. These join the cluster over time, subject to various interactions/rules.

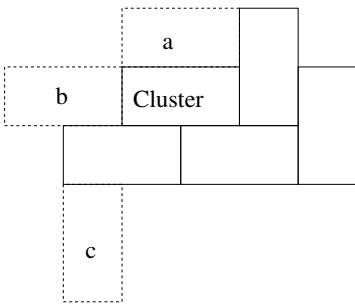

**Figure 2.** The possible "next" domino could be placed in many positions. Three examples are shown. In position "a", it would have three units of perimeter in common with the cluster, "b" two units and "c" one.

The basis of the model is that at very low simulation temperature, the need to maximise contacting perimeter, $p_C$, will dominate behaviour. As $T$ increases, the likelihood of adding a "non-optimal" domino increases. This then allows vacancies to be incorporated into the cluster. At very high temperature, the only constraint on the domino to be added is that it contact the cluster. In all cases, only neighbours above and below, left and right are considered; no diagonal interactions, for example.

In outline, the steps are these:

1. Find all empty squares that are adjacent to the cluster.
2. Find all possible dominoes that could be added to the cluster.
3. Work out the maximum possible contacting perimeter for a new domino. Call this $p_{max}$. In the process, work out perimeters for all possible dominoes. Call these the $p_i$.
4. Randomly choose any one of the possibilities.

5.  Calculate its deficit compared to $p_{\max}$, $\Delta_i = p_{\max} - p_i$.

6.  Generate a random number, $r$, on interval zero to one. If $e^{\frac{-\Delta_i}{T}} > r$, then keep the chosen domino. If not, return to Step 4.

7.  Iterate until some number of steps or some other condition (like meeting with the edge of the simulation space) occurs.

A domino ("A") that has its origin on, say, $x$, and partner square on $x + 1$ is effectively the same as one ("B") that has origin on $x + 1$ and partner on $x$; the algorithm removes duplicates before a domino is chosen at random. A more complex model might allow dominoes to possess some polarisation, such that the two squares were not exactly equivalent. In such a case, A and B would not be duplicates.

For a domino with $p_i = p_{\max}$, $\Delta_i = 0$. If such a domino is chosen at random from all of the possibilities, it will always be accepted. The larger the value of $\Delta_i$ for a given domino, the larger the exponent and the greater the likelihood that it will not be accepted for adding to the cluster (see Step 6 above). An increase in $T$ reduces the exponent, making acceptance more likely. Thus, higher $T$ allows acceptance of non-optimal dominoes.

It is important to note that the rule to maximise the contact perimeter is a local rule, which makes sense if the system is thought of as a simple model of the chemistry of crystallisation or aggregation. A new domino does not "know" how far it is from the centre of the cluster, but it is reasonable to suggest that there is a bonding energy per unit length that means a longer $p_C$ gives a stronger interaction.

If a domino has $p_C$ units of perimeter in common with the cluster, it will cover $p_C$ units of the cluster perimeter if added. In addition, those $p_C$ units of the domino perimeter will not be added to the cluster perimeter.

Hence, since a domino has six units of perimeter, the additional cluster perimeter, $p_{\mathrm{add}}$ will be:

$$p_{\mathrm{add}} = 6 - 2p_C \tag{1}$$

such that if there were an empty, enclosed domino site in the cluster, like a $2 \times 1$ hole, then sticking a domino in it would reduce the cluster perimeter by six units. Therefore, this local rule acts to minimise the cluster perimeter, keeping in mind that lengths are measured using a "city block" metric, effectively, since there are no diagonals.

## 3. Results

*3.1. T = 0*

When simulation temperature is zero, the only randomness comes from which of the maximally contacting dominoes will be chosen to add to the cluster. It is impossible to add a non-optimal domino. A typical example of a resulting cluster is shown in Figure 3.

A movie of the growth of the cluster for $T = 0$ is included with the deposited material. The cluster shows four, or two depending on the definition, domains. Within a domain, the system is highly ordered, although there are stacking faults, such that consecutive layers may slip relative to each other by half a domino length. Much of the growth is deterministic. When there is a step or kink in one side of the cluster (see domino *A* in Figure 4), that is the only place where $p_C = 3$ is possible. Thus, there is deterministic, highly-ordered growth along that edge until the edge is complete. Then, there are multiple sites where $p_C = 2$ is possible, and one of these is randomly chosen. That then determines which edge will grow next, and so on. Before domino *A* was added to the cluster in Figure 4, $p_{\max} = 2$, and this was true for many possible dominoes. Once had *A* been added, there was only one position where a new domino can obtain $p_C = 3$, and so positioning of *B* was non-random, given *A*. Had *A* been in the centre of the side, there would have been two possible positions for *B*, but in either case, the end result is the building of a line of dominoes along that side of the cluster. This gives the layer-by-layer growth observed in more complicated crystal growth models [15,16].

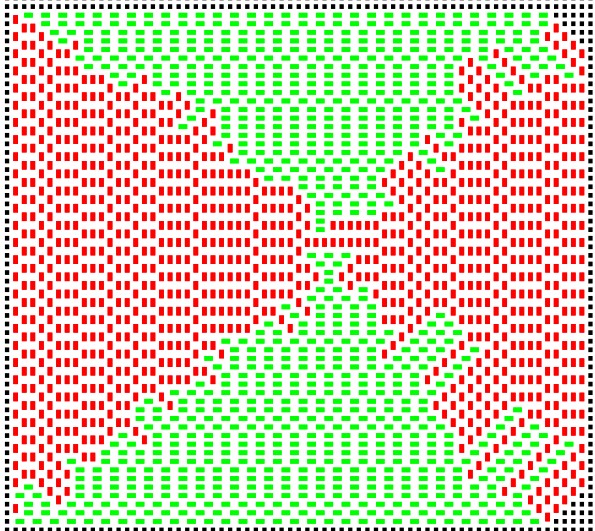

**Figure 3.** The cluster formed subject to maximising the contacting perimeter ($T = 0$). Note the lack of vacancies. This illustration uses the first 2000 dominoes ($N_D = 2000$). Red dominoes are vertical, green horizontal. Empty sites are in black.

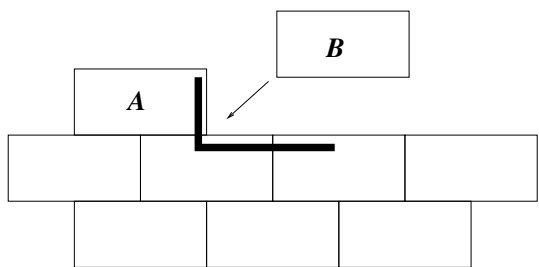

**Figure 4.** When domino *A* is added to the cluster, there is only one place where domino *B* can obtain $p_C = 3$; the kink, indicated by a heavy black line.

### 3.2. T > 0

The maximum possible value for $p_C$ is six, and $T$ is measured on this scale. As $T$ increases, the cluster evolves in two key ways. First, the domains begin to mix, such that eventually, the domain structure breaks down; secondly, vacancies start to appear. The former phenomenon begins to occur at lower temperatures than the latter. At $T = 0.1$, the first "inclusions" appear, and at $T \sim 0.16$, they are beginning to cluster (Figure 5).

The dotted box in Figure 5b shows a region entirely limited to one of the domains. By calculating:

$$\Delta N = \frac{N_V}{(N_V + N_H)} \tag{2}$$

where $N_H$ is the number of horizontal dominoes and $N_V$ the vertical, a measure of the persistence of the domain is obtained. Figure 6 plots $\Delta N$ as a function of $T$. The curved black line on the figure is an exponential fit of the form:

$$\Delta N = A e^{\frac{-(T-B)}{C}} + 0.5 \tag{3}$$

and is added purely to illustrate the convergence of $\Delta N$ towards 0.5 (the black horizontal line). Comparing the scatter of points with the difference between the exponential curve and the $\Delta N = 0.5$ line, it is apparent that the domains are well mixed by $T \sim 0.8$. Indeed, if the fall in $\Delta N$ is modelled as a linear function valid on the domain $0.18 \leq T \leq 0.8$, the dashed line suggests $\Delta N = 0.5$ at $T \sim 0.8$.

The exponential model gives better fit statistics, but uses more parameters, so both merely act as guides to the behaviour and suggest $N_V = N_H$ at around $T = 1$, for this region.

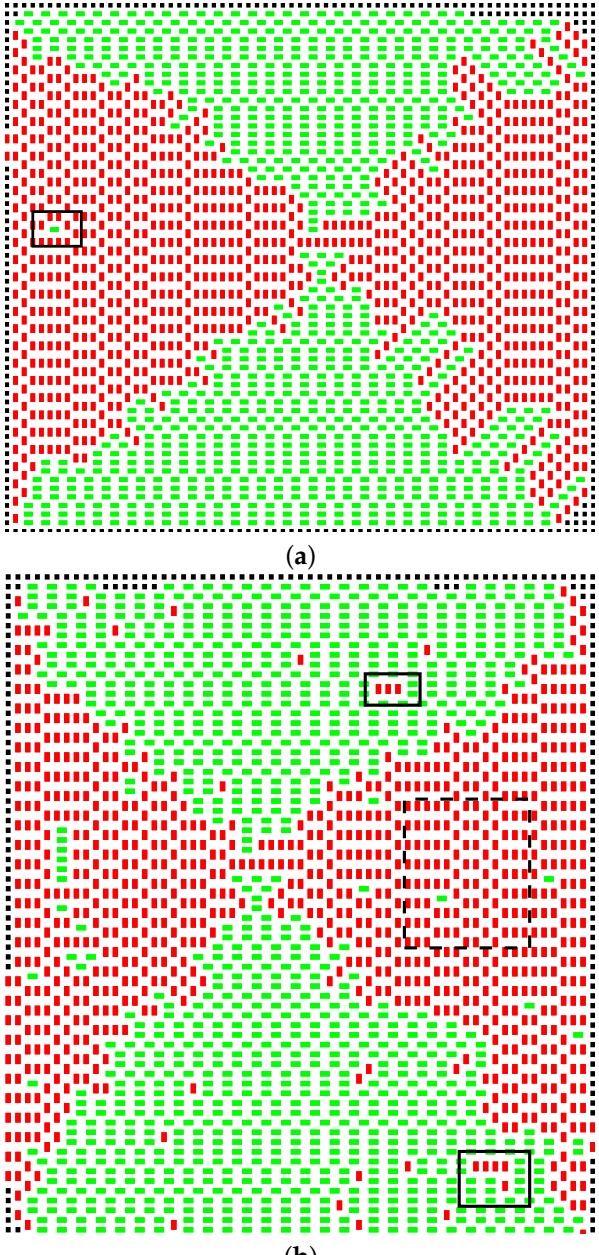

**Figure 5.** (**a**) $T = 0.10$ and (**b**) $T = 0.16$. In (**a**), an inclusion of a green (horizontal domino) is indicated by the black box within the red (vertical dominoes) domain. In (**b**), these inclusions are common enough that they are beginning to cluster and can no longer be considered as isolated defects, but as small domains. The effect grows more pronounced with increasing $T$. Note that as yet, there are no vacancies.

On average, a parallel domino will possess a $p_C$ bigger by one unit of perimeter than a perpendicular domino, as a direct result of their $2 \times 1$ dimensions (the parallel side is one unit longer than the perpendicular side). This means that for a given temperature, the probability of accepting a perpendicular domino is less than that of accepting a parallel one, and the two are related by:

$$P_{\text{perp}} = P_{\text{para}}e^{1/T}. \tag{4}$$

It is clear that the $e^{1/T}$ term will approach unity (meaning perpendicular and parallel additions are equally likely) as $T$ gets very large. This is as expected, as there is no symmetry-breaking interaction in this very simple system.

In the finite box used to generate Figure 6, the populations become within uncertainty of being equal at finite $T$, as noted above.

The breakdown of the flat-edged domain structure is somewhat akin to the roughening transition that occurs in crystal growth as conditions are varied [17–19] and is continuous, as would be expected for such a simple model. By very large temperatures ($T \sim 10$), the vacancy fraction reaches approximately 0.1. The constraint that a new domino must contact the cluster prevents very high vacancy densities.

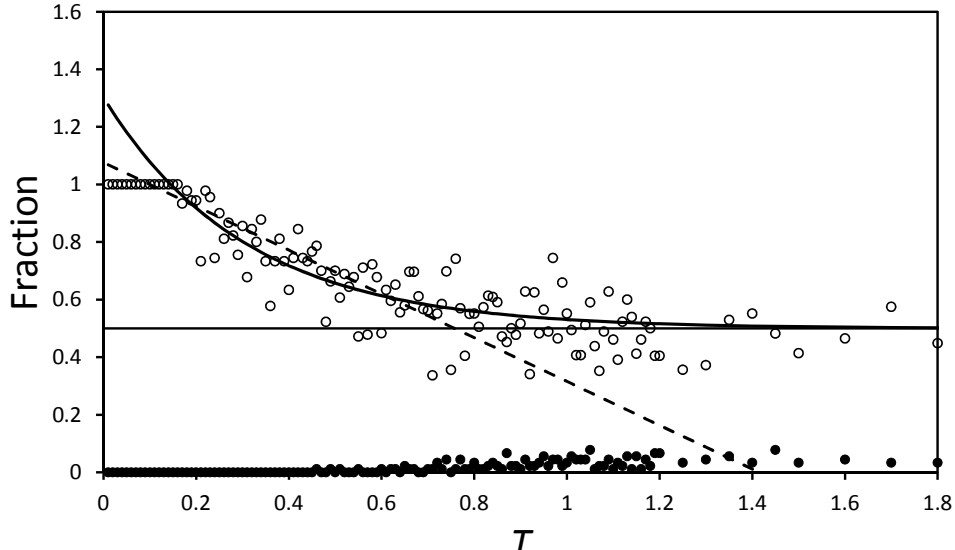

**Figure 6.** The open circles plot $\Delta N$ (see Equation (2)) against $T$. Because a small box (90 lattice points) was used, the data are noisy, but the trend is clear. The solid curving line is a fit to an exponential function and the dashed to a simple linear function. The horizontal line indicates $\Delta N = 0.5$. The fraction of vacancies in the box as a function of $T$ is plotted as closed circles.

The first vacancy appears at around $T \sim 0.3$. At around $T = 0.6$, there are typically about 20 to 30 vacancies across a simulation the size of Figure 3, though the number does fluctuate. At $T \sim 10$, the growth is extremely random, such that vacancies occupy around 10% of the sites within the cluster, and $2 \times 1$ vacancy clusters are present (Figure 7). A domino positioned to fill such a pair of vacancies would be very energetically favourable; it would have a a contacting perimeter of six; but the sites would have to be filled by random chance, and this is unlikely. In a real example of crystallisation, such sites would be inaccessible to the new molecules.

Note that at this high temperature, the growth is erratic, very different from the systematic, square growth pattern observed at low temperatures. This can be best observed in the movie in the deposited material.

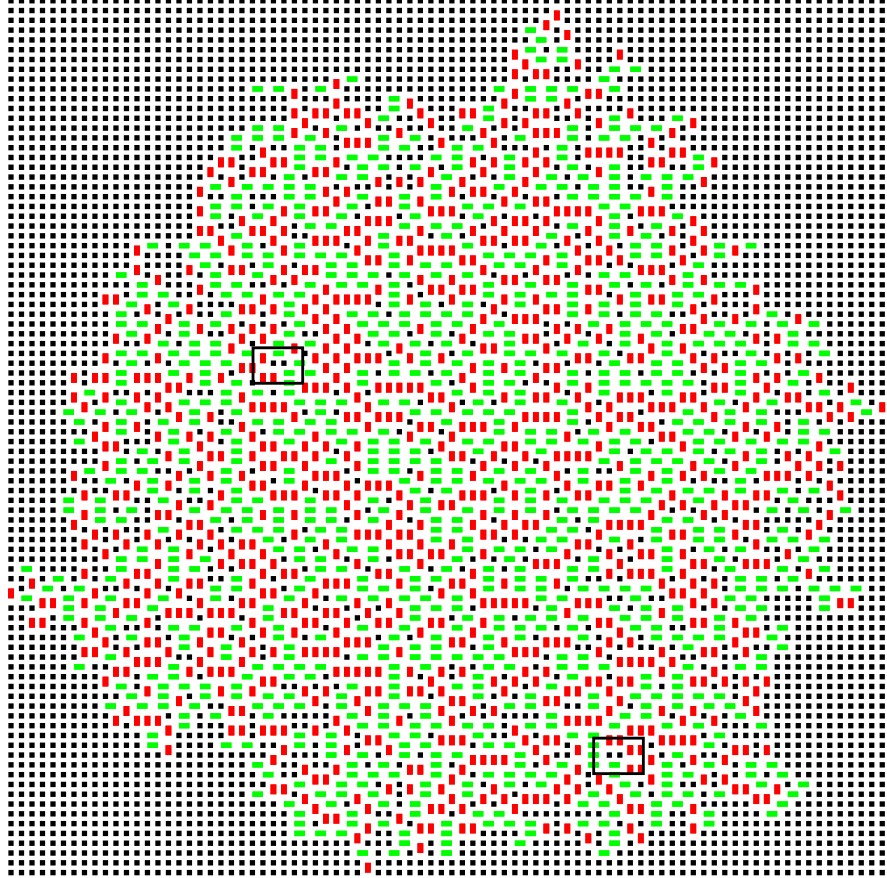

**Figure 7.** $T = 10$ showing the vacancies as black squares amidst the coloured ones. The black boxes highlight enclosed $1 \times 2$ vacancies where a domino could sit. Note that the edges are now very ragged, and growth is irregular and shows no evidence of the original squarer growth pattern; compare for example with the plots in Figure 5.

## 4. Materials and Methods

Simulations were run on a Linux-based workstation. The GFortran (version 4.7.2) compiler was used to compile the code. Both the OS and the compiler are freely available. The code, including any subroutines, is included in the Supplementary Material and is freely available for use and modification.

## 5. Conclusions

Very simple tilings can show a wide range of phenomena, including clustering, surface roughening, inclusions and vacancies, despite simple rules that can be implemented quickly. With further factors, such as different domino dimensions, polarisation of dominoes (where a domino becomes like a polar dimer, a dipole), multiple nucleation sites and modelling of applied fields (which can apply a torque to a dipole and prefer certain orientations), the possibility of modelling a very wide range of phenomena arises.

The local potential used prefers dominoes with the maximum possible "contact area" with the existing cluster (akin to a surface energy minimisation). When temperature is low and the potential dominates, a phase that shows just a few large, highly-ordered domains without vacancies is obtained. As $T$ increases, inclusions of the other orientation are found within domains. Further increase in $T$ allows vacancies to occur and the domain structure to break down, until at very high temperatures, domino orientation is essentially random, and the only constraint is that a new domino must contact the existing cluster.

One reason for doing this work is that such very simple models have not been as widely explored as might be expected, although of course, other workers have used tilings in various contexts [3,10,12]. Much work is either heavily mathematical [20,21] or statistical [22]. The work here is less realistic in its complexities than these studies, but it is hoped that in compensation, it allows conceptually simple exploration of the growth dimer-by-dimer, something not directly tackled by the more complex approaches, and is intended to be complementary to them. The attempt to develop a very simple model based on local interactions will, it is hoped, lead to insights into how the properties of the dimers give rise to the types of large clusters that we see, and do so in a very simple and visual manner that can be tailored to different situations and complexities.

**Supplementary Materials:** The following are available online at http://www.mdpi.com/2410-3896/2/2/15/s1: Video V1: Movie_of_cluster_growth_T=0.mpg; Video V2: Movie_of_cluster_growth_T=10.mpg; Program P1: dom2016G2.f.

**Acknowledgments:** The author thanks colleagues in the School of Physical Environmental and Mathematical Sciences at the University of New South Wales, Canberra, for their encouragement and support, including the provision of computing resources.

**Conflicts of Interest:** The author declares no conflict of interest.

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
