# Peer review of "Exploring Cluster Growth Using a Simple Domino Tiling"

_condensedmatter, doi:10.3390/condmat2020015_

Reviewer 1 Report

The paper can be accepted with minor revisions. The present paper adresses the study of clustering of 2 × 1 dimers (dominoes) subject to simple interactions and temperature, i.e., exploring how the cluster evolves with different parameters. In particular, the temperature effects on the overall shape, surface roughening, number of vacancies, are discussed. This is a very simple modelling approach (e.g., without considering thermodynamic properties), however the code presented in the supplemental material can be extended to more complex situations. The paper is concise and clearly discussed, however, the introduction should be improved introducing similar studies in the literature, emphasing the improvements made here.  

There are some phrases not so readable which should be improved as follows: 

- In line 64, please rewrite the following phrase: "In such a case the two would not be duplicates"

- In line 65, it is confusing: "Since for a domino with pi = pmax ∆i = 0,..." This could be rephrased.

- Please, make the following phrase more readable: "If a domino has pC units of perimeter in common with the cluster, then adding that domino means that pC units of the new domino’s perimeter are not added to the cluster perimeter and at the same time pC units of cluster perimeter have been covered"

Its understandable but not so clear. Its a very large phrase.

Author Response

The author thanks the reviewers for sharpening and improving the expression of the manuscript. Their comments are very useful and their advice has been taken. If possible, please pass along my thanks and respects.

---------------------------------------------------------------

Reviewer 1:

Comments and Suggestions for Authors

The paper can be accepted with minor revisions. The present paper addresses the study of clustering of 2 × 1 dimers (dominoes) subject to simple interactions and temperature, i.e., exploring how the cluster evolves with different parameters. In particular, the temperature effects on the overall shape, surface roughening, number of vacancies, are discussed. This is a very simple modelling approach (e.g., without considering thermodynamic properties), however the code presented in the supplemental material can be extended to more complex situations. The paper is concise and clearly discussed, however, the introduction should be improved introducing similar studies in the literature, emphasing the improvements made here.

Response: One reason for doing this work is that I could not find such simple models in the literature. Other workers have used tilings in various contexts, but the work on dimers seems to mostly come from a thermodynamic perspective that does not allow exploration of the dimer-by-dimer growth, but deals in probabilities and overall averages. I was trying to begin to develop a very simple model based more on local interactions that would ultimately give some insight into how the properties of the dimers give rise to the large clusters that we see, and do so in a very simple and visual manner. Hence, I have added such a paragraph to the conclusions. New citations have been added.

There are some phrases not so readable which should be improved as follows: In line 64, please rewrite the following phrase: "In such a case the two would not be duplicates"

Response: I take it that the problem here was whether I was talking about squares or whole dominoes, and I can see the ambiguity. I have rephrased the paragraph to remove ambiguity.

- In line 65, it is confusing: "Since for a domino with pi = pmax ∆i = 0,..." This could be rephrased.

Response: Rephrased.

- Please, make the following phrase more readable: "If a domino has pC units of perimeter in common with the cluster, then adding that domino means that pC units of the new domino’s perimeter are not added to the cluster perimeter and at the same time pC units of cluster perimeter have been covered"

Its understandable but not so clear. Its a very large phrase.

Response: Rephrased.

Reviewer 2 Report

The manuscript „Exploring cluster growth using a simple domino tiling” by Goossens D. J. concerns tiling of the  plane with dominoes subjecting a simple statistical/geometrical rule with one free parameter (‘temperature’). Numerical simulations show that clustering changes with increase of the ‘temperature’ from the ordered pattern to an irregular one. I agree with the Author that simple toy models are useful and interesting because they can explain some behaviors of complex systems (the manuscript regards to the subject of growth of crystals).

The manuscript is well written and contains new results, therefore it may be published in Condensed Matter. However, some improvements (listed below) should be done.

The Author should stress more clearly (in Introduction or in Conclusions) what is new in his manuscript regarding cited papers on the similar subject.

Some analytical considerations (stochastic analysis) on this toy model would be very welcomed by readers  - for example: an analytical explanation of the observation that at about T = 1 two lines cross (in Fig. 6).

In the first sentence of Conclusions the Author suggests that his model can show phase transitions. However, the simulations do not show the phase transition – I suggest to      remove “phase transitions” from the sentence.  

Author Response

The author thanks the reviewers for sharpening and improving the expression of the manuscript. Their comments are very useful and their advice has been taken.

---------------------------------------------------------------
The manuscript is well written and contains new results, therefore it may be published in Condensed Matter. However, some improvements (listed below) should be done.

The Author should stress more clearly (in Introduction or in Conclusions) what is new in his manuscript regarding cited papers on the similar subject.

Response: One reason for doing this work is that I could not find such simple models in the literature. Other workers have used tilings in various contexts, but the work on dimers seems to mostly come from a thermodynamic perspective that does not allow exploration of the dimer-by-dimer growth, but deals in probabilities and overall averages. I was trying to begin to develop a very simple model based more on local interactions that would ultimately give some insight into how the properties of the dimers give rise to the large clusters that we see, and do so in a very simple and visual manner. Hence, I have added such a paragraph to the conclusions. New citations have been added.

Some analytical considerations (stochastic analysis) on this toy model would be very welcomed by readers  - for example: an analytical explanation of the observation that at about T = 1 two lines cross (in Fig. 6).

Response: The discussion of this figure has been recast heavily. It includes some comments on (very simple!) analytical results.

In the first sentence of Conclusions the Author suggests that his model can show phase transitions. However, the simulations do not show the phase transition – I suggest to remove “phase transitions” from the sentence.

Response: Rephrased.

Round  2

Reviewer 2 Report

The manuscript has been improved according to my suggestions.